# Recent Progress on Biological Activity of Amaryllidaceae and Further Isoquinoline Alkaloids in Connection with Alzheimer’s Disease

**DOI:** 10.3390/molecules26175240

**Published:** 2021-08-29

**Authors:** Lucie Cahlíková, Rudolf Vrabec, Filip Pidaný, Rozálie Peřinová, Negar Maafi, Abdullah Al Mamun, Aneta Ritomská, Viriyanata Wijaya, Gerald Blunden

**Affiliations:** 1ADINACO Research Group, Department of Pharmaceutical Botany, Faculty of Pharmacy, Charles University, Heyrovskeho 1203, 500 05 Hradec Kralove, Czech Republic; pidany@faf.cuni.cz (F.P.); perinovr@faf.cuni.cz (R.P.); negarm@faf.cuni.cz (N.M.); almamuna@faf.cuni.cz (A.A.M.); ritomska@faf.cuni.cz (A.R.); wijayav@faf.cuni.cz (V.W.); 2Department of Pharmacognosy, Faculty of Pharmacy, Charles University, Heyrovskeho 1203, 500 05 Hradec Kralove, Czech Republic; vrabecr@faf.cuni.cz; 3School of Pharmacy and Biomedical Sciences, University of Portsmouth, Portsmouth, Hampshire P01 2DT, UK; gblunden10@gmail.com

**Keywords:** isoquinoline alkaloids, Alzheimer’s disease, acetylcholinesterase, butyrylcholinesterase, prolyl oligopeptidase, monoaminooxidase, neuroprotective activity, docking study

## Abstract

Alzheimer’s disease (AD) is a progressive age-related neurodegenerative disease recognized as the most common form of dementia among elderly people. Due to the fact that the exact pathogenesis of AD still remains to be fully elucidated, the treatment is only symptomatic and available drugs are not able to modify AD progression. Considering the increase in life expectancy worldwide, AD rates are predicted to increase enormously, and thus the search for new AD drugs is urgently needed. Due to their complex nitrogen-containing structures, alkaloids are considered to be promising candidates for use in the treatment of AD. Since the introduction of galanthamine as an antidementia drug in 2001, Amaryllidaceae alkaloids (AAs) and further isoquinoline alkaloids (IAs) have been one of the most studied groups of alkaloids. In the last few years, several compounds of new structure types have been isolated and evaluated for their biological activity connected with AD. The present review aims to comprehensively summarize recent progress on AAs and IAs since 2010 up to June 2021 as potential drugs for the treatment of AD.

## 1. Introduction

Alzheimer’s disease (AD), a multifactorial neurodegenerative disease that affects the elderly population, is the most common form of dementia that is generally diagnosed in individuals over the age of 65 years, and the one with the strongest societal impact with regard to incidence, prevalence, mortality rate, and cost of care [1,2,3]. The prevalence of AD is growing rapidly, in line with an increase in the age of the population. AD currently affects over 50 million people worldwide and is estimated to nearly triple by 2050 [4]. The exact cause of AD is constantly under debate and review. Several hypotheses have been put forward, including cholinergic, amyloid beta (Aβ), hyperphosphorylation of τ-protein, calcium dyshomeostasis, and oxidative stress [5,6]. Pathological changes in the brain of a patient with AD can occur more than twenty years before clinical symptoms are apparent [7]. From a neuropathological point of view, the most pronounced selective cholinergic neuronal loss is accompanied by the formation of intracellular neurofibrillary tangles (NFTs) composed of hyperphosphorylated τ-protein and extracellular senile plaques (SPs) formed by β-amyloid peptide (Aβ) [8]. SPs are extracellular deposits composed of various peptide fragments (Aβ_(1–40)_ and Aβ_(1–42)_) derived from the amyloid precursor protein (APP) [9]. These proteins generate deposits in specific areas of the brain and are considered critical factors for memory loss and cognitive impairment in AD patients [10].

Acetylcholinesterase (AChE, EC 3.1.1.7) is a serine-protease that plays a pivotal role in cholinergic transmission in the central nervous system and at the neuromuscular junctions (NMJ). The physiological function of AChE is to hydrolyze acetylcholine (ACh) to afford choline (Chol) and acetic acid, thus playing a key role in cholinergic neurotransmission within the autonomic and somatic nervous system [11]. AChE is mainly expressed in nervous tissue, neuromuscular junctions, plasma and red blood cells [2,12]. In the human brain, there are two structural types of AChE: type G4 (tetramer, highly prevalent) and monomer G1 (in minor amount). The proportion of G1 (monomer) is significantly increased in AD [13].

Similar to AChE, another cholinesterase, butyrylcholinesterase (BuChE, EC 3.1.1.8) can also inactivate ACh. Whilst, under normal conditions, BuChE is responsible for 20% of the total brain cholinesterase activity, in the later stages of AD the activity of BuChE increases to 40–90% in certain brain regions. This dramatic switch between the AChE/BuChE ratio implies a change from a supportive role to a leading role for BuChE in hydrolyzing the excess of ACh [14]. The two enzymes have different abilities to hydrolyze substrates. These differences are probably caused by changes in the arrangement of amino acids in the aromatic cavity. Altered expression of AChE in the brain of patients with AD suggests that AChE activity increases at the periphery of the amyloid plaque (around the Aβ plaques) and Aβ may actually affect the levels of AChE. It has been found that different forms of AChE in the brain and cerebrospinal fluid of patients with AD are changed in connection with abnormal glycosylation [15]. While the role of AChE in neurodegenerative diseases is relatively well known, that of BuChE is still not completely clarified. This pseudocholinesterase does not have natural substrates in the organism [13]. Some studies have shown that BuChE may indirectly contribute to the pathogenesis of type 2 diabetes mellitus (T2DM) by causing insulin resistance [16]. Sharing the pathological similarities between AD and T2DM (enhanced amyloid beta aggregation, abnormal levels of cholinesterase) gave rise to the hypothesis that cholinesterase inhibitors (ChEIs) can potentially treat both these disorders. Although ChEIs afford mostly a symptomatic response to AD patients, the development of new ChEIs (e.g., multifunctional ligands, selective BuChE inhibitors) remains of interest to treat neurodegenerative diseases. ChEIs may also be used as a treatment for an autoimmune disease named myasthenia gravis, by targeting nicotinic acetylcholine receptors at the neuromuscular junction, and as a prophylactic treatment against organophosphorus nerve agent poisoning [17,18].

Excitatory glutamatergic neurotransmission via the *N*-methyl-d-aspartate receptor (NMDAR) is critical for synaptic plasticity and the survival of neurons. Glutamate is the most abundant excitatory neurotransmitter in the mammalian CNS. It is extensively distributed in the CNS, where it is almost exclusively located intracellularly [19]. However, excessive NMDAR activity causes excitotoxicity and promotes cell death, underlying a potential mechanism of neurodegeneration that occurs in AD. The glutamatergic hypothesis postulates that the progressive cognitive decline seen in AD patients is due to neuronal cell death caused by the overactivation of NMDA receptors and the subsequent pathological increase in intracellular calcium. Several studies have shown that both soluble Aβ oligomers and τ-protein can interact with several glutamate signaling proteins, such as NMDA receptors and proteins involved in glutamate uptake and recycling, leading to glutamate excitotoxicity [20]. Several current AD treatments target glutamatergic signaling, including memantine, which is an FDA-approved low-affinity NMDA receptor antagonist.

As mentioned above, one of the neuropathological characteristics of AD is the presence of NFTs, in which the formation enzyme glycogen synthase kinase-3β (GSK-3β, EC 2.7.11.26) plays an important role. GSK-3β is a multifunctional serine/threonine kinase, which is recognized as a key component of multiple signaling pathways [2]. It plays a crucial role in cellular functions, including cell-cycle regulation, differentiation, and proliferation. More recently, GSK-3β has been found to play a crucial role in neurodegeneration in general and AD in particular [21]. In AD, the overactivity and/or overexpression of GSK-3β accounts for memory impairment, tau hyperphosphorylation, increased β-amyloid production and local plaque-associated microglial-mediated inflammatory responses, which are hallmarks of the disease [22]. Since the phosphorylation of τ-proteins is primarily dependent on GSK-3β and cyclin-dependent kinase 5 (CDK5, EC 2.7.11.22) [23], inhibition of GSK-3β and CDK5 is accepted as a promising strategy for the treatment of AD [24].

In pathological processes such as neurodegeneration and depression, low levels of monoamine neurotransmitters have been found. Studies have shown that activated monoamine oxidase (MAO) in the brain of patients with AD is a biomarker for the disease, and it is presumed to induce overproduction of hydroxyl radicals in the brain triggering a biochemical cascade connected to the deposition of Aβ plaques [25,26]. MAO exists as two isoenzymes in humans—MAO-A and MAO-B—which are distinct due to different amino acid sequences, three-dimensional structures, distribution in organs and tissue, inhibitor sensitivities and substrate specificity. Activated MAO-B leads to cognitive dysfunction, destroys cholinergic neurons, causes disorder of the cholinergic system and contributes to the formation of amyloid plaques [27]. MAO inhibitors have neuroprotective effects related to oxidative stress, which are desirable properties for the development of multi-target drugs for AD.

In recent years, the enzyme prolyl oligopeptidase (POP, EC 3.4.21.26) has gained importance as a target for the treatment of schizophrenia (SZ), bipolar affective disorder (BD) and cognitive disturbances, such as those present in Alzheimer’s disease (AD). POP is a serine protease of the S9 protein family that cleaves peptide bonds at the carboxyl end of proline [28]. POP is primarily expressed in the hippocampus, hypothalamus, amygdala, cortex and striatum regions of the brain [29]. It has been discussed that reduced POP activity correlated to the tau pathology and severity of AD [30]. Moreover, POP is associated with learning and memory because it degrades several neuro-peptides; thus, inhibition of POP can impose an interesting alternative for AD treatment [31].

The current treatment, based on the cholinergic hypothesis, is only symptomatic and mainly involves the restoration of ACh levels through AChE inhibition [11]. Three AChE inhibitors, namely, donepezil, galanthamine, and rivastigmine, and one fixed-combination of donepezil and memantine (approved in 2014) are currently used as the main therapeutic option for AD treatment [32,33]. These available drugs are marketed for mild to severe stages of AD. They are able to alleviate cognitive and behavioral symptoms for only a limited period (few months after the start of treatment) [32].

In June 2021, a new AD drug named aducanumab, sold under the brand name Aduhelm, was approved in the USA by the FDA. Aducanumab, a human IgG1 monoclonal antibody, is the first drug with a putative disease-modifying mechanism for the treatment of this devastating disorder, namely, the removal of β-amyloid (or Aβ) plaques from the brain [34]. The decision was highly controversial and led to the resignation of three FDA advisers because of the absence of evidence that the drug is effective, as clinical trials gave conflicting results on its effectiveness [35,36].

Natural products represent an important source of clinical drugs, especially for their structural diversity and a wide range of biological activities [37]. Alkaloids are, without a doubt, the most intriguing templates of natural origin [38]. Given the fact that Alzheimer’s disease has become an important topic in recent years, and a large number of papers have been published on this subject, this review covers the recent progress on Amaryllidaceae alkaloids and further isoquinoline alkaloids, and their biological activity related to AD published between 2010 and 2021. Previous results can be found in the following review articles [39,40,41,42]. Recently, a detailed review on cholinesterase inhibitors from marine organisms has been published [41].

## 2. Isolation of Alkaloids from Plant Material for Biological Evaluation

The critical step in the biological evaluation of AAs is their isolation from plant materials, and thus much of the activity has been directed towards the development of optimum methods for the isolation of alkaloids and alkaloid mixtures. Because the alkaloids usually occur in plants as salts of organic plant acids and inorganic acids, together with complex mixtures of water-soluble compounds, it is often a problem to remove all of these non-alkaloidal compounds during isolation and purification [43,44,45]. Most alkaloids are insoluble in water and more or less soluble in such organic solvents as ether, chloroform, dichloromethane, and ethyl acetate, whereas their salts have just the opposite solubility characteristics [46].

The general methods of extraction and isolation of alkaloids include the separation of them from the main bulk of the non-alkaloidal substances with solvents such as methanol and ethanol; in some studies, *n*-butanol has also been used [44,47]. The crude concentrated extract is often subjected to successive extractions with ether, ethyl acetate, chloroform or dichloromethane using the acid-base properties of alkaloids. When the crude extract is exposed to an alkaline medium, the alkaloidal salts are readily converted to the corresponding alkaloid bases to form a concentrated alkaloidal extract. These concentrated alkaloidal extracts are subsequently separated by either silica gel or aluminum oxide column chromatography (CC), flash chromatography (FC), preparative high-performance liquid chromatography (HPLC), preparative thin layer chromatography (TLC), and crystallization [43,46]. CC is used especially when a higher amount of plant material is available for the isolation of AAs (normally over 1 kg). Because of the possible formation of isolation artifacts of alkaloids (like *N*-chloromethyl-, methoxy-, ethoxy- and butoxy-derivatives), it is necessary to specify the used solvents within all steps of the phytochemical study to identify if the isolated compound is really a natural product [48,49,50]. Alcohols can react with a carboxylic group to form esters, and with hemiacetals to form acetals. As a result of the interaction of alcoholic solvents (e.g., methanol and ethanol) with hydroxy groups, methoxy and ethoxy functions can be detected within isolated products. Within Amaryllidaceae alkaloids (AAs), butoxy-derivatives have also been described when *n*-butanol has been used within the phytochemical study [49,51]. Halogen-containing solvents are very unstable, because of their direct interaction with molecules, and because of the presence of contaminants as well [52]. It has been reported that chloroform can directly react with alkaloids, as in the case of AAs and protoberberine-type isoquinoline alkaloids [53]. Furthermore, chloroform can accelerate the decomposition processes of compounds, especially when the solutions of fractions and subfractions are not stored in the dark [52].

## 3. Amaryllidaceae Alkaloids

AAs are one of the most studied groups of secondary metabolites in connection with AD. Chemically, they represent a large group of isoquinoline alkaloids, which are produced exclusively by plants of the Amaryllidaceae family and are reported as a separate group of alkaloids. These alkaloids are derived from the aromatic acids phenylalanine and tyrosine, which are used to produce key intermediates in the biosynthesis of the AA 4-*O*-methylnorbelladine [54]. This compound subsequently undergoes different types of intramolecular oxidative couplings, which leads to the formation of the basic structural types of AAs: galanthamine, lycorine, haemanthamine, crinine, homolycorine, tazettine, and montanine.

The biological activity of Amaryllidaceae plants is connected with the presence of AAs, which demonstrate a wide range of biological activities, including antitumor, antibacterial, antioxidant, antiparasitic, antifungal, anti-inflammatory and insect antifeedant effects, as well as acetylcholinesterase and butyrylcholinesterase inhibitory activities [47,55,56]. The most known AA is galanthamine, originally isolated from *Galanthus woronowii* Losinsk. [57], which is a reversible, competitive acetylcholinesterase inhibitor. Additionally, galanthamine binds to the allosteric sites of nicotinic receptors, which causes a conformational change [58]. This allosteric modulation increases the nicotinic receptor’s response to acetylcholine. The activation of presynaptic nicotinic receptors increases the release of acetylcholine, further increasing its availability [59]. 

Galanthamine was approved by the FDA (Food and Drug Administration) in 2001 for the treatment of mild to moderate stages of AD. In the early 21st century, it was reported that the acetylcholinesterase activity of AAs is mainly associated with the galanthamine- and lycorine- structural types [60], but in the last decade, new structural types of AAs have been identified as interesting cholinesterase inhibitors (Table 1). 

Two new structural types of AAs, named narcikachnine- and carltonine-type, have been isolated in the last five years from various Amaryllidaceae plants. The former combines in its structure both the galanthamine- and galanthindole-type. So far, five AAs of this structural type have been isolated from *Zephyranthes citrina*, *Narcissus pseudonarcissus* cv. Carton, *N*. *pseudonarcissus* cv. Dutch Master, and *N. poeticus* cv. Pink Parasol [61,62,63,64], and four of them were screened for *h*AChE/*h*BuChE inhibition potential (Table 1, Figure 1 and Figure 2). 

Before the results of the inhibition potency of newly isolated and tested alkaloids are discussed, it is worth mentioning that, for the determination of cholinesterase inhibition potential, it is important that the source of the used enzymes is reported, e.g., electric eel acetylcholinesterase (*Ee*AChE), human acetylcholinesterase *h*AChE, mouse brain acetylcholinesterase, equine serum BuChE (*Eq*BuChE), and human BuChE (*h*BuChE). The use of different types of enzyme can result in dramatic differences in the obtained IC_50_ values. This can be demonstrated with acetylcaranine, which showed interesting activity against *Ee*AChE (IC_50_ = 11.7 ± 0.7 μM) [65]. When *h*AChE was used for the study, acetylcaranine was evaluated as a weak inhibitor (IC_50_ = 443.7 ± 62.4 μM) [66]. This phenomenon has also been described recently for 1-*O*-acetyllycorine [66], reported as a strong *Ee*AChE inhibitor (IC_50_ = 0.96 ± 0.04 μM) [67,68], but inactive against *h*AChE (IC_50_ = > 1000 μM) [66]. In Table 1 and Table 2, the source of enzyme for each IC_50_ value is stated.

The recently isolated narcibaduliine from *N. pseudonarcissus* cv. Carton demonstrated a similar inhibitory activity towards both cholinesterases, with IC_50_ values of 3.29 ± 0.73 μM for *h*AChE, and 3.44 ± 0.02 μM for *h*BuChE [64]. The best *h*BuChE inhibition activity within the narcikachnine-type AAs has been demonstrated by narcieliine (IC_50_ = 1.3 ± 0.3 μM) [63]. The only structural difference from narciabduliine was found in a C-3′,C-4′-substitution in the galanthindole moiety (Figure 1; differences marked in red). Narcieliine was also subjected to enzyme kinetics analysis to determine the kinetics of *h*AChE/*h*BuChE inhibition. Enzyme velocity curves were recorded at several concentrations of narcieliine, and analysis of them revealed a competitive type of inhibition for *h*AChE and a mixed type of inhibition for *h*BuChE. Due to the isolation of only a low amount of narciabduliine, the type of inhibition of this alkaloid has not been studied. To determine the critical structure aspects responsible for the *h*AChE/*h*BuChE activities of narcieliine and narciabduliine, molecular modeling studies were applied [63,64]. In the case of narciabduliine, the top-scored position in the cavity of *h*AChE revealed that the galanthamine moiety is bound to the catalytic anionic site (CAS), whereas the galanthindole core is lodged peripherally. Interaction of narciabduliine with the oxyanion hole, formed by Gly120-122 residues, is mediated via hydrogen bonds to both the hydroxyl group of the cyclohex-2-en-1-ol moiety and the methoxy group of the phenyl ring. Moreover, the hydroxyl group seems to stand aside from the Ser203 residue, and rotation of Ser203 to a plausible interaction with the hydroxyl moiety may occur. The galanthindole moiety is sandwiched between Tyr337 and Tyr341 by parallel π–π stacking, hampering the hydrogen-bond donor contact between the hydroxyl group of Tyr337 to nitrogen from an azepine ring in narciabduliine. In general, the overall topology of narciabduliine in *h*AChE shares a high similarity to that of galanthamine, but the galanthindole core allowed additional spanning of narciabduliine into the peripheral anionic site (PAS) of the enzyme. The best scoring pose of narciabduliine in the active site of *h*BuChE indicated that the ligand adopted an inverse accommodation to that observed for the *h*AChE complex. The galanthindole moiety is buried inside the enzyme’s cavity, whereas the galanthamine core of narciabduliine is situated distally. The critical interaction for galanthindole is distorted π–π stacking between the 1-methyl-2,3-dihydro-1*H*-indole moiety of narciabduliine and Trp82, a hydrogen bond between the oxygen of the methoxy group and hydrogen at the 1-methyl-2,3-dihydro-1*H*-indole moiety, two hydrogen bonds between the phenolic hydroxyl functionality of narciabduliine and glycine residues 116 and 117, and plausible hydrogen contact to Ser198 from the catalytic triad. The galanthamine moiety seems to contribute less to the overall ligand anchoring, forming hydrophobic contacts with Tyr332 and Gln119 only (Figure 2) [64]. 

Similar interactions have been observed for narcieliine, which expands the catalytic anionic site (CAS) of *h*AChE entirely, with several apparent interactions. The “galanthamine” core is lodged centrally, and the “galanthindole” part is protruding out of the gorge, contacting several other amino acid residues. A closer view of the interaction between narcieliine and *h*BuChE revealed that the galanthindole moiety is anchored in the vicinity of the catalytic triad, but not involved in direct contact with this part of the enzyme [63].

Narcieliine was also tested for its POP inhibition potential, reaching the same magnitude of inhibition (IC_50_ = 163 ± 13 μM) as that of berberine, which is recognized as a natural POP inhibitor (IC_50_ = 142 ± 21 μM) [75]. The highest POP inhibition potential within the isolated narcikachnine type alkaloids has been shown by narcimatuline, isolated from *N. poeticus* cv. Dutch Master, with an IC_50_ value of 29.1 ± 1.0 μM (Table 1, Figure 1). This alkaloid also showed interesting *h*BuChE activity (IC_50_ = 5.9 ± 0.2 μM) and has been identified as a mild GSK-3β inhibitor (IC_50_ = 20.7 ± 2.4 μM) [62]. 

Since the prediction of CNS availability is critical for the development of drugs for neurodegenerative diseases, the logBB values of the most active compounds have been calculated. This value predicts the logarithmic ratio between the concentration of the compound in the brain (C*_brain_*) and blood (C*_blood_*) [76]. The obtained logBB values for narciabduliine and narcieliine (Figure 1) indicated that both alkaloids should be able to reach the target area in the CNS. Moreover, the PAMPA-BBB screening method was also applied for narcieliine to predict passive diffusion through biological membranes, employing a brain lipid porcine membrane [77,78]. The obtained result for the in vitro permeability of narcieliine (*Pe* = 14.1 ± 1.0 10^−6^ cm s^−1^) indicated that it can cross the BBB by passive diffusion. Narciabduliine was unable to be studied for in vitro permeability due to the isolation of only a low amount of this alkaloid [64].

Taken together, the narcikachnine type is an interesting group of AAs with promising biological activities in connection with the potential treatment of AD. Unfortunately, these compounds are present in plants only in low concentration; thus, the development of a synthetic route is of interest.

The second newly discovered structural type of AAs, the carltonine-type, contains in its structure a belladine moiety combined with a lycosinine fragment (Figure 3). In the *h*AChE assay, all members of the carltonine-type displayed marginal inhibition potency (Table 1). On the other hand, these compounds showed promising inhibition activity towards *h*BuChE (Table 1). Carltonine A and carltonine B displayed IC_50_ values in the nanomolar range (*h*BuChE IC_50_ = 910 nM, and 31 nM, respectively). From the structural perspective, both AAs are endowed with the same core structure, differing only in the substitution at positions C-5′and C-6′, respectively (Figure 3). The presence of a 1,3-dioxalane ring in carltonine B compared with its opened dimethoxybenzene analogue carltonine A is plausibly responsible for the almost 30 times drop in *h*BuChE inhibition activity (Figure 3; the differences marked in blue). The last carltonine-type alkaloid, carltonine C, contains in its structure an additional lycosinine fragment (Figure 3); this additional part of the structure is connected with a dramatic decline in the *h*BuChE inhibition activity (IC_50_ = 14.8 ± 1.1 μM) [56].

Molecular modeling studies were applied to reveal fundamental interactions and to identify structure details responsible for the dramatic drop in *h*BuChE inhibition activity. The in silico observations deduced that the higher inhibition potency of carltonine B over carltonine A might be attributed to the presence of the 2*H*-1,3-benzodioxole moiety for its “extended” aromatic properties that, especially in the *pseudo*-enantiomer, broaden the range of hydrophobic interactions between ligand and enzyme [56]. 

Due to the isolated amounts, only carltonine A has been studied for POP inhibition potency. It showed activity in the same range as that of berberine (Table 1). The significant *h*BuChE inhibition activity of alkaloids of this structural type and the fact that these alkaloids are present in plants only in trace amounts (from 30 kg of fresh bulbs of *N*. *pseudonarcissus* cv. Carlton, 70 mg of carltonine A, 6 mg of carltonine B, and 7 mg of carltonine C were isolated) mean that the total synthesis of these compounds is a future challenge for medicinal chemistry. 

Close derivatives with a norbelladine framework (*N*-benzyl-2-phenylethan-1-amine congeners) have been developed by Carmona-Viglianco et al. [79] and screened for their AChE/BuChE potency (Figure 4). Generally, these derivatives yielded moderate to weak inhibition potency against both cholinesterases with IC_50_ values above 10 µM. The structure of the best AChE/BuChE inhibitor within this study (**1**) is displayed in Figure 3, together with belladine-type AAs. The latest study published by Al Mamun et al. [80] reported synthesis of compounds with the same fundamental unit, but the developed compounds possessed different substitution patterns within the A-ring (Figure 3). Seven compounds of the twenty synthesized possessed a strong and selective *h*BuChE inhibition profile, with IC_50_ values below 1 μM [80]. The most potent one showed nanomolar range activity with an IC_50_ value of 72 nM and an excellent selectivity pattern over AChE, reaching almost 1400 (Figure 3). The top-ranked compound (**2**) was further studied by enzyme kinetics, along with in silico techniques, to reveal the mode of inhibition. The prediction of CNS availability estimates that all synthesized compounds within the survey can pass through the blood–brain barrier (BBB), as disclosed by the BBB score. These synthetic compounds, inspired by the norbelladine structural type of AAs, display an interesting structure scaffold for further development of selective *h*BuChE inhibitors [80]. 

A phytochemical study of bulbs of *Lycoris longituba* yielded thirteen AAs that have been screened for their neuroprotective, and electric eel AChE (*Ee*AChE) inhibition activity [70]. The natural existence of isolated alkaloids with a *N*-chloromethyl moiety in the structure, as in *N*-(chloromethyl)galanthamine and *N*-(chloromethyl)lycoramine isolated from this plant must be first reinvestigated, because when halogenated solvents are used during the isolation process, this can result in the formation of *N*-chloromethyl alkaloid derivative artifacts (Figure 4) [48,49]. The best *Ee*AChE inhibitors belong to the galanthamine type (e.g., *N*-norgalanthamine (IC_50_ = 2.76 ± 0.65 μM) and 11β-hydroxygalanthamine (IC_50_ = 3.04 ± 0.61 μM)). Interesting activity was also obtained for AAs of further structural types (e.g., narciclassine type AA *N*-methylcrinasiadine (IC_50_ = 4.23 ± 1.13 μM; Table 1)) [70]. Some of the isolated alkaloids also showed promising neuroprotective effects against CoCl_2_, H_2_O_2_, and Aβ_25–35_-induced neuronal cell death in dopaminergic neuroblastoma SH-SY5Y cells. Incartine, trisphaeridine, *N-*(chloromethyl)galanthamine, sanguinine, *O*-demethyllycoramine, and deoxypretazettine showed significant neuroprotective effects against all three injury models [70]. Lycolongirine C and *N*-methylcrinasiadine exhibited significant neuroprotective activities against H_2_O_2_ and Aβ_25–35_-induced cell death [70]. Homolycorine- and galanthamine-type AAs isolated from the bulbs of *L. aurea* showed significant neuroprotective effects against CoCl_2_-induced SH-SY5Y cell injury, while *N-*norgalanthamine, *N-*(chloromethyl)galanthamine, *N*-(chloromethyl)lycoramine, 2α-hydroxy-6-*O*-methyloduline, 2α-hydroxy-6-*O-n*-butyloduline, *O-n-*butyllycorenine, *O-*methyllycorenine-*N*-oxide and pluviine showed obvious neuroprotective effects against H_2_O_2_-induced SH-SY5Y cell death [51]. The same models have been used for the evaluation of neuroprotective alkaloids isolated from *L. sprengeri*. Two alkaloids, namely, *O-*methyllycorenine and hippadine, exhibited significant neuroprotective effects against H_2_O_2_-induced SH-SY5Y cell death; lycosprenine, *O-*methyllycorenine, and tortuosine showed obvious neuroprotective effects against CoCl_2_-induced SH-SY5Y cell injury [81]. The results obtained within the mentioned studies indicate that AAs of homolycorine-, lycorine-, and galanthamine-type may have potential for further development as neuroprotective compounds. The cited studies are one of the few that have addressed the neuroprotective activity of individual AAs. Most of the studies dealing with the neuroprotective activity of AAs have used complex alkaloidal extracts accompanied by GC/MS analysis [82,83,84]. However, studies performed on mixtures are not able to identify which compound is responsible for the studied activity. Moreover, synergistic effects can occur. It is highly unlikely that a complex mixture will be used for the treatment of AD, and thus studies of individual compounds are required.

Under pathological conditions, such as in AD, the imbalance of the neurotransmitter leads to excitotoxic processes. There is an increase in the intraneuronal concentrations of Ca^+^, which causes mitochondrial dysfunction, the activation of proteases and the accumulation of reactive oxygen species (ROS) and reactive nitrogen species (RNS) [85]. The brain is particularly vulnerable to damage caused by ROS and RNS, due to the high consumption of oxygen and its high lipid content [86]. Galanthamine is able to stabilize free radicals due to the presence of the enol group in its structure [87]. It is well known that molecules with phenolic groups have a greater capacity to stabilize ROS and RNS. On the other hand, it has been shown that these groups are inefficient when crossing the BBB [88], a pharmacokinetic limitation for drugs which should reach the CNS area, as in the treatment of AD. Recently, the antioxidant activity of several fractions of alkaloids isolated from Amaryllidaceae species has been studied using a model of glutamate (Glu) excitotoxicity in rat cortical neurons [89]. The study showed a possible neuroprotective mode of action of the alkaloidal fractions of *Eucharis bonplandii* bulbs, *E. caucana* bulbs and *Clivia miniata* leaves due to the presence of phenol and enol groups in the structures of the AAs. Additionally, the pair of free electrons on the N is spatially close to hydroxyl groups, which benefits the cleavage of this group and consequently the stabilization of the generated O· [89].

The alkaloidal extract of *Hippeastrum reticulatum* bulbs (AHR) has been studied for its ameliorating effects on memory and cognitive dysfunction in a scopolamine-induced rodent model of AD using a Y-maze, the novel object recognition test (NORT), and the Morris water maze (MWM) test [90]. Sixty Swiss male mice, divided into six groups, received samples for 15 days. The normal group received saline and the scopolamine-treated group scopolamine (1.5 mg/kg, intraperitoneal injection). The test group received AHR (5, 10 and 15 mg/kg, per os) and the positive control group donepezil (5 mg/kg, per os), administered 1 h before the test; scopolamine was injected 30 min prior to testing. The results of in vivo studies showed that AHR (15 mg/kg) significantly increased spontaneous alternation performance in the Y-maze test, and significantly increased the time spent exploring the novel object compared with the scopolamine-treated group. In the MWM test, the administration of AHR at doses of 10 and 15 mg/kg significantly decreased escape latency and swimming distance to the platform, which indicates learning and spatial memory abilities of the mice [90].

Two standardized alkaloidal extracts of *Zephyranthes carinata* (bulbs and leaves) were evaluated for their in vitro and in vivo neuroprotective potential. In order to select the most promising extract for in vivo study in a triple transgenic mouse model of AD (3xTg-AD), an in vitro excitotoxicity model was used. The extract of bulbs of *Z. carinata* (BZC) protected neurons against a Glu-mediated toxic stimulus in vitro, and consequent in vivo experiments showed the effects of the intraperitoneal administration of BZC extract (10 mg/kg) every 12 h for 1 month on learning and spatial memory of 3xTg-AD (18 months old) [91]. Histopathologically, a significant reduction in tauopathy and astrogliosis was observed, as well as a significant decrease in the proinflammatory marker COX-2, which indicates the potential of BZC alkaloids for further biological evaluation.

Further AAs, isolated and tested in the last decade, with interesting biological activity related to AD are summarized in Table 1 and Figure 4 and Figure 5.

## 4. Further Isoquinoline Alkaloids

Isoquinoline alkaloids (IAs) belong to one of the most complex families of plant alkaloids. They are widely distributed in plants coming from the families Papaveraceae, Rutaceae, Berberidaceae, Menispermaceae, Ranunculaceae, and Annonaceae [92]. Several species of Magnoliaceae and Convolvulaceae are also rich in these alkaloids and have been intensively investigated for their various biological activities [93]. Based on different degrees of oxygenation, intramolecular rearrangements, distribution, and the presence of additional rings connected to the main system, they may be divided into eight subgroups: benzylisoquinoline, aporphine, protoberberine, benzo[*c*]phenanthridine, protopine, phthalideisoquinoline, morphinan, and emetine alkaloids. The largest and most studied group among the listed subgroups is protoberberines, the activity of which on cholinesterases has been evaluated by numerous authors, mainly regarding the quaternary compounds berberine and palmatine [94,95,96]. In the last decade, several new IAs with interesting biological activity towards AD have been isolated.

The detailed phytochemical study of the alkaloidal extract of the root bark of *Berberis vulgaris* (Berberidaceae) led to the isolation of ten isoquinoline alkaloids. Compounds isolated in sufficient amounts were evaluated for their in vitro *h*AChE, *h*BuChE, POP and GSK-3β inhibitory activities [97]. Significant *h*BuChE inhibition potential was demonstrated by the newly isolated bisbenzylisoquinoline alkaloids aromoline (IC_50_ = 0.82 ± 0.10 μM), and berbostrejdine (IC_50_ = 6.9 ± 1.0 μM). Both alkaloids were only weak *h*AChE inhibitors. On the other hand, aromoline showed only slightly lower POP inhibition potency than the used standard, berberine (Table 2). The balanced *h*BuChE and POP inhibitory abilities of aromoline initiated its screening for GSK-3β inhibition activity at a concentration of 10 μM. Unfortunately, the assayed compound showed only weak inhibition potency (Table 2). Kinetic studies of aromoline indicated that it acts as a mixed inhibitor of *h*BuChE. To identify structural determinants responsible for the ligand binding, a molecular modeling simulation exploiting the crystal structure of *h*BuChE (PDB ID: 4BDS) was carried out [98]. Aromoline was seen to be lodged near all of the catalytic triad residues establishing parallel *π–π* interactions of the phenolic moiety with His438. This part of the molecule of aromoline was implicated in a conventional hydrogen bond with Ser198. Additionally, Glu197 showed van der Waals forces with aromoline, and Tyr332, enabling the formation of T-shaped *π–π* stacking with one of the 1,2,3,4-tetrahydroisoquinoline moieties. 

The whole plant of *Berberis vulgaris* is generally considered a rich source of the quaternary protoberberine alkaloids berberine and palmatine, which have been identified as AChE inhibitors. The biological activities of these compounds in relation to AD have been recently summarized in various articles and detailed information can be found in the following reviews [38,99,100,101,102,103,104,105,106,107,108].

Extensive phytochemical study of whole plants of *Corydalis mucronifera* (Papaveraceae) yielded previously undescribed IAs, named mucroniferanines A-M, together with 26 known IAs [109,110]. The isolated alkaloids were evaluated using *Ee*AChE [110] and selected IAs also with equine serum BuChE [109]. Of the newly isolated compounds, the most promising activities have been shown by the quaternary protoberberine alkaloid mucroniferanine H, with IC_50_ values of 2.31 ± 0.20 μM for *Ee*AChE, and 36.71 ± 1.12 μM for *Eq*BuChE. Interesting inhibition potency has also been obtained for further mucroniferanines (Table 2). The type of inhibition and identification of structural determinants responsible for the ligand binding have not been determined. 

Twelve IAs were isolated from the whole plant of *Corydalis saxicola* (Papaveraceae) and screened for their *Ee*AChE and *Eq*BuChE inhibition potential [96]. Six of the isolated IAs, including berberine and palmatine, showed potent inhibition of *Ee*AChE activity (Table 2). Structure-activity studies demonstrated that the nitro substituent in the structure of protoberberines, such as in 1-nitroapocavidine, increases AChE inhibition potency. All of the studied alkaloids were inactive against *Eq*BuChE (IC_50_ > 200 μM) [96].

**Table 2 molecules-26-05240-t002:** Biological activities of further IAs, isolated and tested in connection with potential treatment of Alzheimer’s disease, from 2010 (alkaloids only included with an IC_50_ value ≤ 30 μM for AChE or BuChE; an IC_50_ ≤ 200 μM for POP; all results for MAO-A and inhibition of Aβ-aggregation are included).

Isoquinoline Alkaloid	Structural-Type	Plant Source	IC_50_ AChE (μM)	IC_50_ BuChE (μM)	IC_50_ POP (μM)	IC_50_ MAO-A (μM)	IC_50_ Aβ (μM)	Ref.
avicine	benzophenanthridine	*Zanthoxylum rigidum*	0.15 ± 0.01 ^a^ 0.52 ± 0.05 ^b^	0.88 ± 0.08 ^c^	n.s	0.41 ± 0.02	5.56 ± 0.94	[111]
nitidine	benzophenanthridine	*Zanthoxylum rigidum*	0.65 ± 0.09 ^a^ 1.25 ± 0.09 ^b^	5.73 ± 0.60 ^c^	n.s	1.89 ± 0.17	1.89 ± 0.40	[111]
6-ethoxydihydrochelerythrine	benzophenanthridine	*Chelidonium majus*	0.83 ± 0.04 ^b^	4.20 ± 0.19 ^d^	n.s.	n.s.	n.s.	[112]
chelerythrine	benzophenanthridine	*Chelidonium majus*	1.54 ± 0.07 ^b^ 3.78 ± 0.15 ^a^	6.33 ± 0.93 ^c^ 10.34 ± 0.24 ^d^	n.s.	0.55 ± 0.04	4.20 ± 0.43	[113,114]
sanguinarine	benzophenanthridine	*Corydalis saxicola*	1.93 ± 0.01 ^a^	>200 ^c^	n.s.	n.s.	n.s.	[96]
mucroniferanine H (racemic mixture)	protoberberine	*Corydalis mucronifera*	2.31 ± 0.20 ^a^	36.71 ± 1.12 ^c^	n.s.	n.s.	n.s.	[109]
6-ethoxydihydrosanguinarine	benzophenanthridine	*Chelidonium majus*	3.25 ± 0.24 ^b^	4.51 ± 0.31 ^d^	n.s.	n.s.	n.s.	[112]
hydrohydrastinine	simple isoquinoline	*Corydalis mucronifera*	9.13 ± 0.15 ^a^	>100 ^c^	n.s.	n.s.	n.s.	[109]
dehydrocavidine	protoberberine	*Corydalis saxicola*	9.92 ± 0.24 *	>200 *	n.s.	n.s.	n.s.	[96]
mucroniferanine G (diastereoisomeric mixture 1:1)	phthalideisoquinoline	*Corydalis mucronifera*	11.3 ± 0.8 ^a^	n.s.	n.s.	n.s.	n.s.	[110]
mucroniferanine F (diastereosomeric mixture 1:1)	phthalideisoquinoline	*Corydalis mucronifera*	12.2 ± 0.2 ^a^	n.s.	n.s.	n.s.	n.s.	[110]
(+)-canadine	protoberberine	*Corydalis cava*	12.4 ± 0.9 ^b^	>100 ^d^	152.0 ± 12.5	n.s.	n.s.	[115,116,117]
hendersine B	benzylisoquinoline	*Corydalis mucronifera*	14.22 ± 0.34 ^a^	>100 ^c^	n.s.	n.s.	n.s.	[109]
7,8-dihydro-[1,3]-dioxolo[4,5-g]isoquinoline	simple isoquinoline	*Stephania rotunda*	18.4 ± 1.4 ^e^	n.s.	n.s.	n.s.	n.s.	[95]
caryachine	pavinane	*Eschscholzia californica*	19.6 ± 0.4 ^b^	>100 ^d^	n.s.	n.s.	n.s.	[118]
(+)-canadaline	benzylisoquinoline	*Corydalis cava*	20.1 ± 1.1 ^b^	85.2 ± 2.2 ^d^	n.s.	n.s.	n.s.	[115]
remerine	aporphine	*Stephania rotunda*	20.7 ± 1.3 ^e^	n.s.	n.s.	n.s.	n.s.	[95]
chelidonine	benzophenanthridine	*Chelidonium majus*	26.8 ± 1.2 ^b^	31.9 ± 1.4 ^d^	n.s.	n.s.	n.s.	[112]
(−)-mucroniferanine D	phthalideisoquinoline	*Corydalis mucronifera*	28.3 ± 0.4 ^a^	n.s.	n.s.	n.s.	n.s.	[110]
aromoline	bisbenzylisoquinoline	*Berberis vulgaris*	>100 ^b^	0.82 ± 0.10 ^d^	189 ± 32	n.s.	n.s.	[97]
berbostrejdine	bisbenzylisoquinoline	*Berberis vulgaris*	66.0 ± 8.0 ^b^	6.9 ± 1.0 ^d^	n.s.	n.s.	n.s.	[97]
*N*-methylcoclaurine	benzylisoquinoline	*Peumus boldus*	>100 ^b^	15.0 ± 1.4 ^d^	n.s.	n.s.	n.s.	[119]
1-(3-hydroxy-4-methoxybenzyl)-2-methyl-6,7-methylenedioxy-1,2,3,4-tetrahydroisoquinoline	benzylisoquinoline	*Eschscholzia californica*	>100 ^b^	27.8 ± 0.4 ^d^	n.s.	n.s.	n.s.	[118]
sinactine	protoberberine	*Fumaria offcinalis*	>100 ^b^	>100 ^d^	53 ± 2	n.s.	n.s.	[120]
californidine iodide	pavinane	*Eschscholzia californica*	36.7 ± 0.9 ^b^	>100 ^d^	55.6 ± 3.5	n.s.	n.s.	[118,121]
bersavine	bisbenzylisoquinoline	*Berberis vulgaris*	68.0 ± 11.0 ^b^	>100 ^d^	67 ± 6	n.s.	n.s.	[97]
parfumidine	spirobenzylisoquinoline	*Fumaria offcinalis*	>100 ^b^	>100 ^d^	99 ± 5	n.s.	n.s.	[120]
dihydrosanguinarine	benzophenanthridine	*Macleaya cordata, Corydalis mucronifera*	>100 ^b^	n.s.	99.1 ± 7.6	n.s.	n.s.	[110,121]
corypalmine	protoberberine	*Corydalis cava*	>100 ^b^	>100 ^d^	128.0 ± 10.5	n.s.	n.s.	[117,121]
*N*-methyllaurotetanine	aporphine	*Peumus boldus*	>100 ^b^	>100 ^d^	135.0 ± 11.7	n.s.	n.s.	[119,121]
sinoacutine	morphinane	*Peumus boldus*	>100 ^b^	>100 ^d^	143.1 ± 25.4	n.s.	n.s.	[119]
bicuculline	phthalideisoquinoline	*Fumaria offcinalis*	>100 ^b^	>100 ^d^	190 ± 50	n.s.	n.s.	[120]

^a^*Ee*AChE; ^b^*h*AChE; ^c^*Eq*BuChE; ^d^*h*BuChE; ^e^ rat cortex AChE; * origin of used enzymes not specified within this study.

Multi-step chromatography was used for the isolation of alkaloids from the root extract of *Zanthoxylum rigidum* (Rutaceae) [111]. Eight purified IAs were isolated, which were tested for *Ee*AChE, *h*AChE and *Eq*BuChE inhibitory activity, as well as for monoamine oxidase (MAO-A and B) and Aβ aggregation. Two quaternary benzophenanthridine alkaloids, nitidine and avicine, were identified as multi-target candidates. Both alkaloids presented inhibitory activities against both forms of AChE with IC_50_ values in the (sub)micromolar concentration range (Table 2) and showed preferential activity for AChE. Moreover, avicine also demonstrated significant *Eq*BuChE activity (IC_50_ = 0.88 ± 0.08 μM), with an *Eq*BuChE/*h*AChE selectivity index of 1.67, confirming the dual inhibitory character of this compound. Kinetic studies indicated that avicine and nitidine are reversible-mixed inhibitors of both cholinesterases. In the MAO inhibition assay, both alkaloids demonstrated inhibition potency against isoform A of human recombinant monoaminooxidase in micromolar concentrations (Table 2) but were inactive against MAO-B (IC_50_ > 100 μM). In addition, these alkaloids presented moderate Aβ_1-42_ anti-aggregation activity (Table 2).

Seven IAs isolated from *Chelidonium majus* and *Corydalis yanhusuo* were evaluated for their potency to inhibit isoforms of recombinant MAOs. All alkaloids were first screened at a concentration of 20 μM; chelerythrine was selected for the determination of the IC_50_ for MAO-A, and corydaline for MAO-B. Chelerythrine showed similar selective MAO-A inhibition potency (IC_50_ = 0.55 ± 0.042 μM) as avicine, as discussed above [113]. The time-dependence, reversibility and kinetic study of MAO-A inhibition by chelerythrine was also studied. No changes in activity were observed after up to 30 min preincubation; thus, chelerythrine interacts with MAO-A instantaneously and acts as a reversible competitive inhibitor of MAO-A (*K_i_* = 0.22 ± 0.033 μM). Docking simulation implied that Cys323 and Tyr444 of MAO-A are key residues for hydrogen-bond interaction with chelerythrine [113].

For the next study, roots and aerial parts of *Chelidonium majus* (Papaveraceae) were used; eight AAs of different structural types were isolated and screened for *h*AChE/*h*BuChE inhibitory activity [112]. Two compounds, identified as 6-ethoxydihydrosanguinarine and 6-ethoxydihydrochelerythrine, should be recognized as isolation artifacts, as in the presence of common solvents such as MeOH, EtOH and CHCl_3_, the free benzophenanthridine alkaloids (sanguinarine and chelerythrine) easily produce adducts [122]. On the other hand, these artifacts showed balanced cholinesterase inhibition activity, with IC_50_ values lower than 5 μM; 6-ethoxydihydrochelerythrine was the more potent (Table 2). These compounds can be used as templates for further development of antidementia drugs bearing a benzophenanthridine scaffold in the structure.

Column chromatography and preparative TLC of the alkaloid extract of dried tubers of *Corydalis cava* (Papaveraceae) led to the isolation of fifteen IAs belonging to six structural types [117]. Two protoberberine alkaloids, (+)-canadine and (+)-canadaline, were recognized as moderate *h*AChE inhibitors (IC_50_ = 12.4 ± 0.9 and 20.1 ± 1.1 μM, respectively; Table 2). As a part of an ongoing study, (+)-canadine showed POP inhibition potency comparable with that of the used standard, berberine [115].

Repeated column chromatography, preparative TLC, and crystallization led to the isolation of fourteen IAs from the roots and aerial parts of *Eschscholzia californica* (Papaveraceae). Most of the isolated alkaloids were recognized as weak inhibitors of *h*AChE/*h*BuChE. Moderate *h*AChE activity was shown by the pavinane-type alkaloid caryachine (IC_50_ = 19.6 ± 0.4 μM), and moderate *h*BuChE inhibition potency by the newly isolated benzylisoquinoline alkaloid 1-(3-hydroxy-4-methoxybenzyl)-2-methyl-6,7-methylenedioxy-1,2,3,4-tetrahydroisoquinoline (IC_50_ = 27.8 ± 0.4 μM) [118].

Extensive purification of the alkaloidal extract of whole plants of *Fumaria officinalis* (Papaveraceae) afforded twenty alkaloids. Compounds isolated in sufficient amounts have been screened for *h*AChE, *h*BuCh, POP and GSK-3β inhibition activity. Of the studied alkaloids, only parfumidine and sinactine exhibited interesting inhibition potency against POP (IC_50_ = 99 ± 5 and 53 ± 2 μM, respectively) [120]. The other isolated IAs have been considered inactive.

Two benzophenanthridine IAs (sanguinarine, chelerythrine) were evaluated for their potential to inhibit Aβ_1-42_ aggregation [123]. The obtained results indicate that the benzophenanthridine nucleus of both alkaloids is directly involved in the inhibitory effect on Aβ_1-42_ aggregation.

Dauricine, a bisbenzylisoquinoline alkaloid isolated from the rootstock of *Menispermum dauricum* (Menispermaceae) [124], has been studied for its neuroprotective effects in a murine neuroblastoma cell line (N2a) stably transfected with the human Swedish mutant form of amyloid protein precursor (APP), an AD-like cell model [125]. Employing this cell model, which overexpresses APP and hyperphosphorylates tau protein [126], and ELISA and Western blot analysis, it has been revealed that dauricine inhibited APP processing and reduced Aβ accumulation [125].

Another bisbenzylisoquinoline alkaloid, tetrandrine, originally isolated from the dried root *of Stephania tetrandra* [127], attenuated spatial memory impairment and hippocampal inflammation by inhibiting NF-κB activation in a rat model of AD induced by A_β1–42_ [128].

Pronuciferine, a naturally occurring proaporphine alkaloid isolated from *Berberis coletioides* [129], has been investigated for its neuroactive effect against H_2_O_2_-induced apoptosis in SH-SY5Y cells [130]. It did not show any toxicity when SH-SY5Y cells were treated with a concentration of 100 μM or less. The effect of pronuciferine on cell metabolites and brain-derived neurotrophic factor (BDNF) level in SH-SY5Y has been also investigated. Pronuciferine significantly increased the intracellular BDNF protein expression at 10 μM, and thus this alkaloid can be recognized as a neuroactive molecule that might act as a neuroprotective agent to prevent apoptosis in neurodegenerative diseases.

An overview of the IAs isolated and tested in the last decade, with interesting biological activity related to AD, are summarized in Table 2 and Figure 6, Figure 7 and Figure 8.

## 5. Selected Semisynthetic Derivatives of Amaryllidaceae and Isoquinoline Alkaloids as Inspiration for Drug Development for Neurodegenerative Diseases

Since AAs and IAs display promising biological activities and some of them are available from plant material in gram amounts [56,62,73], different semisynthetic derivatives have been developed in the last ten years and screened for biological activities connected with AD.

Two series of aliphatic and aromatic esters of haemanthamine were synthesized by Kohelová et al. [131] and Peřinová et. al. [132]. The free hydroxyl group on C-11 of haemanthamine was esterified with different anhydrides and also acylated with differently substituted benzoyl chlorides affording the corresponding esters [131]. The strongest *h*AChE inhibition capacity within twenty-four haemanthamine derivatives was demonstrated by the 11-*O*-(2-nitrobenzoyl- (**3**), 11-*O*-(3-nitrobenzoyl)- (**4**) and 11-*O*-(2-chlorobenzoyl)- (**5**) derivatives of haemanthamine with IC_50_ values of 9.9 ± 0.5 μM for **3**, 4.0 ± 0.3 μM for **4**, and 13 ± 1 μM for **5** (Figure 9). More interestingly, compounds **3** and **4** displayed a selective *h*AChE inhibition profile with a selectivity index (SI) higher than 10 (Figure 9) [132]. From *h*BuChE data, two derivatives exerted low micromolar inhibition activity, namely, 11-*O*-(2-methoxybenzoyl)haemanthamine (**6**) with an IC_50_ value of 3.3 ± 0.4 μM and 11-*O*-(2-chlorobenzoyl)haemanthamine (**5**) with an IC_50_ value of 5.6 ± 0.6 μM (Figure 9). Moreover, compound **5** also showed interesting *h*AChE inhibition potency and can be considered a dual non-selective cholinesterase inhibitor. The BBB permeability was investigated applying the PAMPA assay to predict CNS availability.The obtained results indicated that all developed derivatives should be centrally active, crossing the BBB by passive diffusion [132].

The same procedure for preparation of semisynthetic derivatives has been used in the case of ambelline, which differs from haemanthamine only in the orientation of the 5,10*b*-ethano bridge and by possessing an additional methoxy group at C-7 (Figure 9). Comparison of the *h*AChE/*h*BuChE inhibition potencies of the analogues with the same substitution related to both ambelline and haemanthamine show the critical role of the absolute configuration of the 5,10*b*-ethano bridge in terms of its biological activity [133]. Within twenty ambeline derivatives, only three of them were identified as *h*AChE inhibitors, namely, 11-*O*-(2-methylbenzoyl)- (**7**), 11-*O*-(2-nitrobenzoyl)- (**8**), and 11-*O*-(4-nitrobenzoyl)ambelline, but these revealed only rather moderate inhibition, with IC_50_ values of 55 ± 4, 54 ± 2, and 48 ± 2 μM, respectively. On the other hand, four aromatic derivatives with either different substitutions on the aromatic ring or with a naphthyl ring exerted strong inhibitory potency against *h*BuChE (IC_50_ < 1 μM). The strongest *h*BuChE inhibition was shown by 11-*O*-(1-naphthoyl)ambelline (**9**), 11-*O*-(2-methylbenzoyl)ambelline (**7**), and 11-*O*-(2-methoxybenzoyl)ambelline (**10**) with IC_50_ values of 0.10 ± 0.01, 0.28 ± 0.02, and 0.43 ± 0.04 μM, respectively (Figure 9). Moreover, the results for **7** and **10** showed a selective inhibition pattern for *h*BuChE with a selectivity index (SI) higher than 100 (Figure 9) [133]. The in vitro results were supported by computational studies predicting plausible binding modes of the strongest inhibitors in the active site of *h*BuChE [133].

As discussed earlier, berberine possesses an interesting multipotent pharmacological profile potentially applicable for AD treatment. Hence, a series of twenty-two berberine derivatives was developed by Sobolová et al. in 2020 and tested in vitro for inhibition potential against *h*AChE/*h*BuChE and POP [134]. The berberine core was substituted at position 9-*O* of its aromatic ring region. All the derivatives displayed *h*AChE inhibition ability in the micromolar to sub-micromolar range. The most active compound, 9-*O*-(α-methylbenzyl)berberine bromide (**11**), displayed inhibitory activity, with an IC_50_ value of 0.79 ± 0.02 μM [134]. In the *h*BuChE assay, all derivatives exerted inhibition activity with IC_50_ values in the micromolar range (1.26–21.5 μM), all being more potent than the parent, berberine (IC_50_ = 47.2 ± 3.3 μM). The top-ranked compound was 9-*O*-(napht-2-yl)berberine bromide (IC_50_ = 1.26 ±0.02 μM) (**12**). All the berberine derivatives were also screened for their POP inhibition activity; 9-*O*-(4-methylbenzyl)berberine bromide (**13**) was the best POP inhibitor with an IC_50_ value of = 10.7 ± 1.2 μM (Figure 9) [134].

## 6. Conclusions and Future Perspectives

The present review demonstrates that, during the last decade, a number of AAs and IAs have been isolated and tested against various targets connected with the potential treatment of AD. Two new structural types of AAs have been isolated, identified and screened for biological activity connected with AD. Compounds of the carltonine-type have been identified as strong selective inhibitors of BuChE, and compounds of the narcikachnine-type showed inhibitory activity against both cholinesterases. Several IAs have been identified as multitarget ligands for AD. Moreover, some of the active compounds have also been investigated through molecular docking studies to identify the sites of interaction between the chemical scaffold and the enzymatic amino acid residues.

As described in Chapter 5, several AAs and IAs can be isolated in gram amounts, allowing for their structure modification, as demonstrated for ambelline, haemanthamine (both AChE/BuChE inactive) and berberine. Thus, the preparation of derivatives of major AAs and IAs may be a new direction in the development of AD drugs. In the case of a low presence of AAs and IAs in plant material, their structures can be used as structural scaffolds for the total synthesis of new compounds, as discussed in the case of molecules structurally inspired by alkaloids isolated from *N. pseudonarcissus* cv. Carlton.

Conclusively, we have shown that AAs and IAs still have promising neuropharmacological potential for further exploration and structure optimalization into multi-target-directed drugs for AD. Nowadays, in vivo and clinical trials are especially needed to be more adequately addressed in order to generate a more coordinated and focused means of developing new AD drugs.

## Figures and Tables

**Figure 1 molecules-26-05240-f001:**
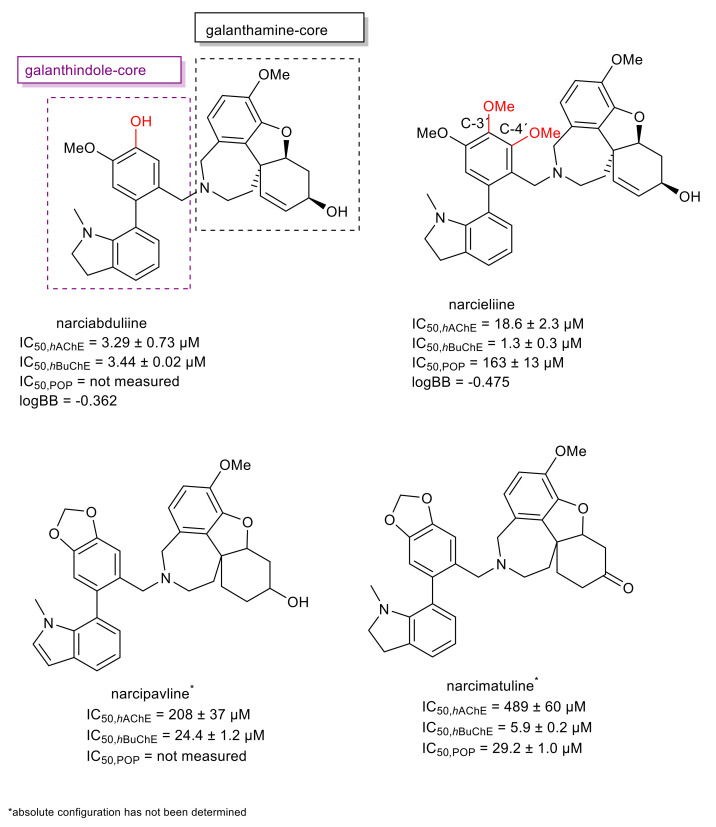
Newly isolated narcikachnine-type AAs together with biological activities connected with AD.

**Figure 2 molecules-26-05240-f002:**
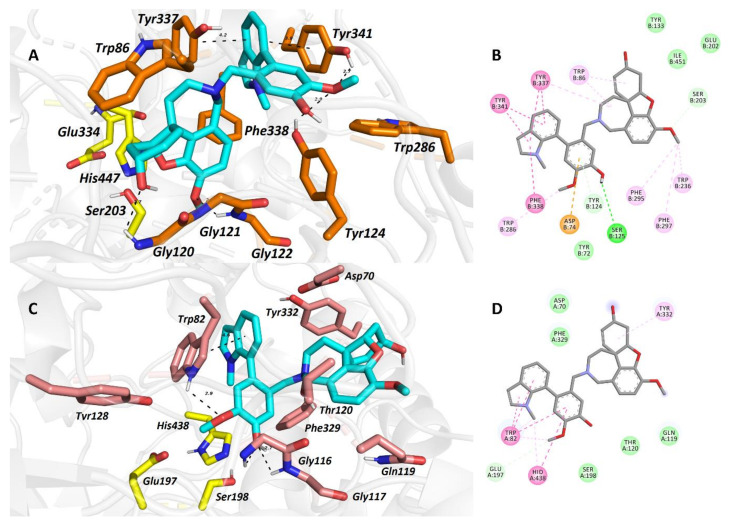
The top-scored docking poses of narciabduliine in *h*AChE ((**A**,**B**); PDB ID: 4EY6) and *h*BuChE ((**C**,**D**); PDB ID: 4BDS) active sites, with important interactions. Spatial orientation for each ligand is presented as three-dimensional (**A**,**C**) and two-dimensional (**B**,**D**) diagrams, respectively (taken from [64]).

**Figure 3 molecules-26-05240-f003:**
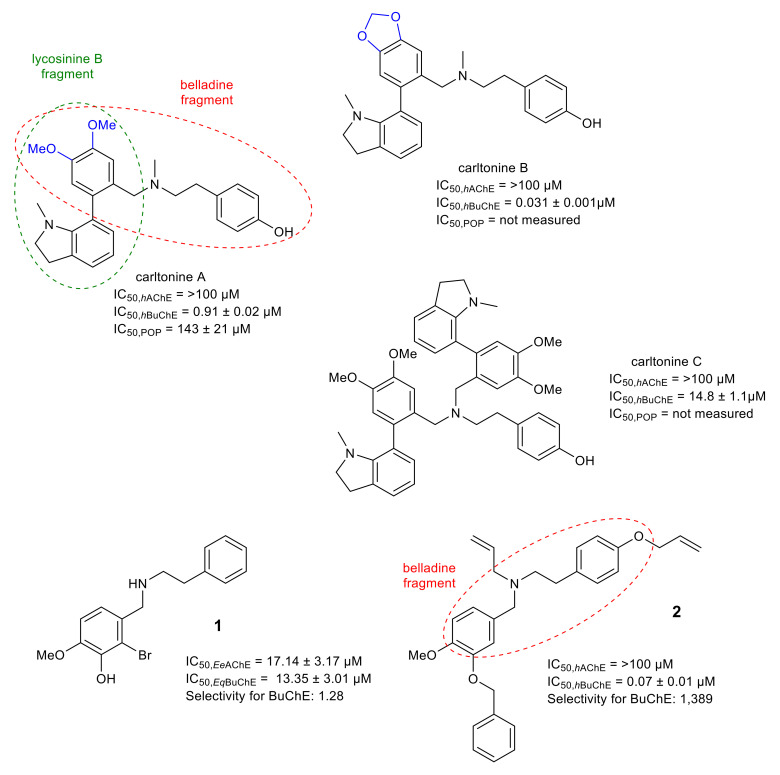
Recently isolated belladine-type Amaryllidaceae alkaloids and the selected synthetic compounds inspired by AAs together with biological activities connected with Alzheimer’s disease.

**Figure 4 molecules-26-05240-f004:**
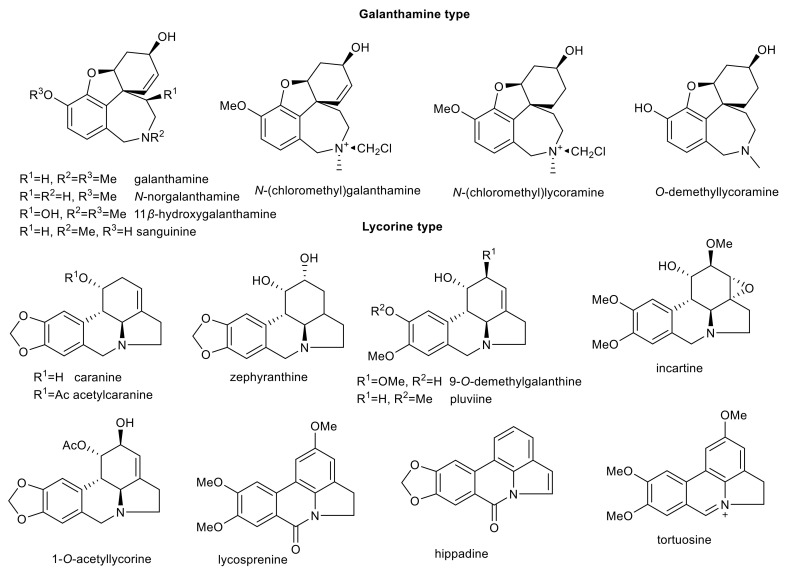
Amaryllidaceae alkaloids of galanthamine- and lycorine-type with interesting biological activity in connection with AD.

**Figure 5 molecules-26-05240-f005:**
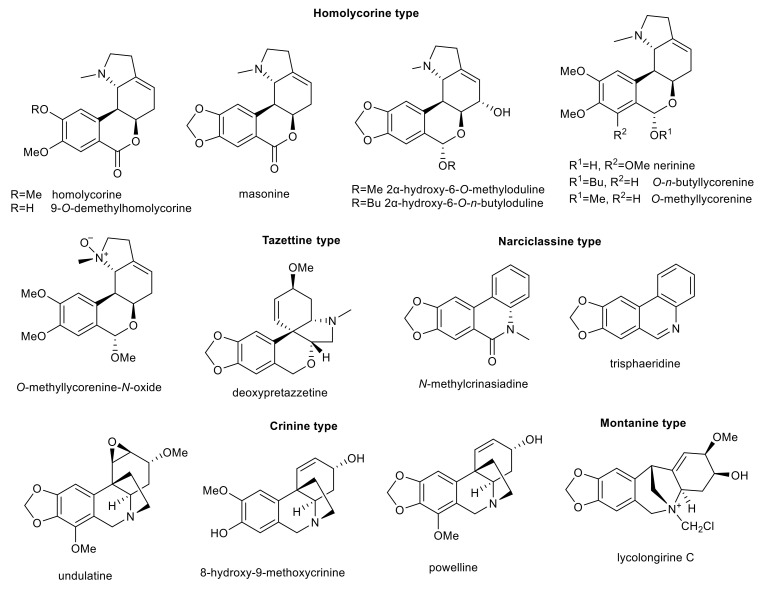
Amaryllidaceae alkaloids of homolycorine-, tazettine-, narciclassine-, crinine, and montanine-type with interesting biological activity in connection with AD.

**Figure 6 molecules-26-05240-f006:**
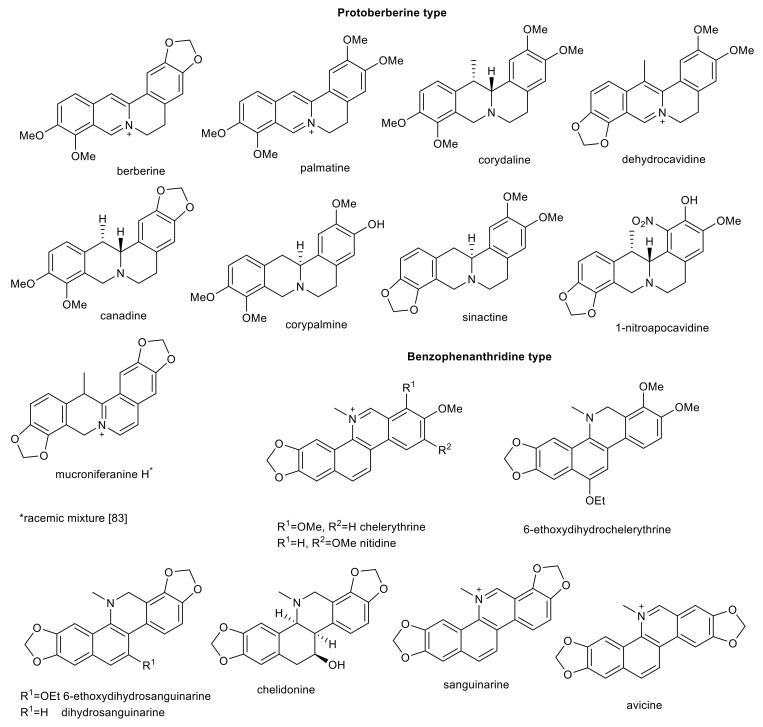
Isoquinoline alkaloids of protoberberine- and benzophenathridine-type with interesting biological activity in connection with AD.

**Figure 7 molecules-26-05240-f007:**
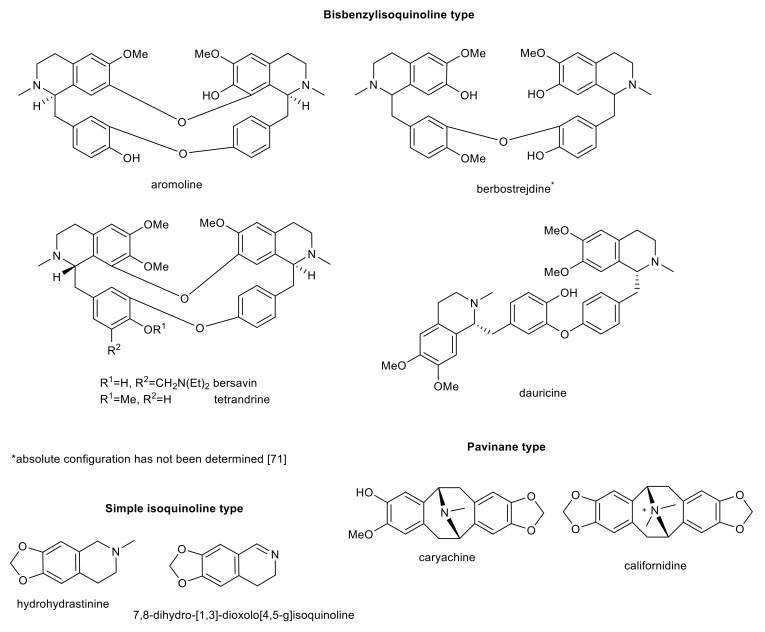
Isoquinoline alkaloids of bisbenzylisoquinoline- and pavinane-types and simple isoquinoline alkaloids with interesting biological activity in connection with AD.

**Figure 8 molecules-26-05240-f008:**
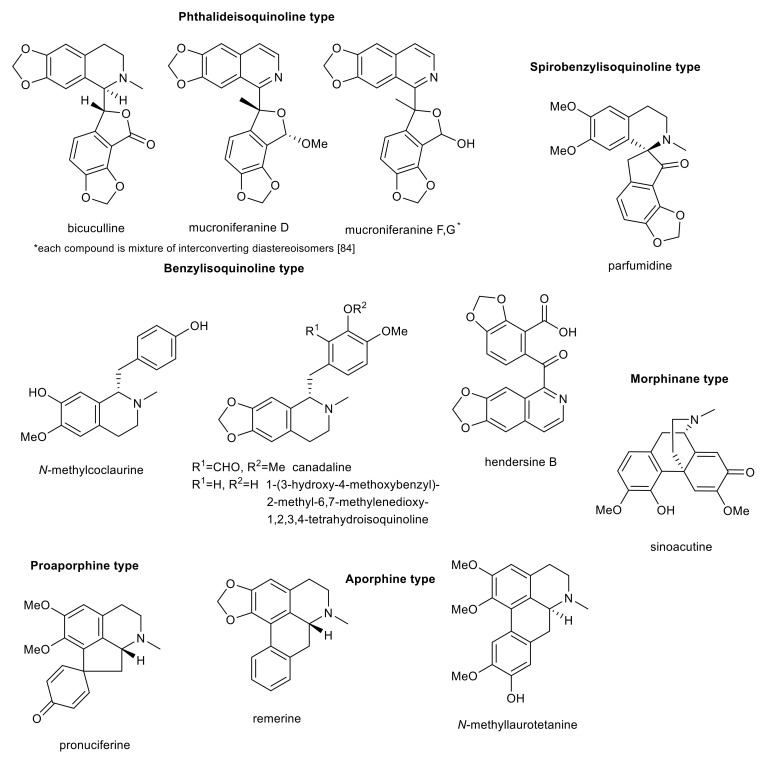
Isoquinoline alkaloids of phthalideisoquinoline-, spirobenzylisoquinoline-, benzylisoquinoline-, morphinane-, proaporphine-, and aporphine-type with interesting biological activity in connection with AD.

**Figure 9 molecules-26-05240-f009:**
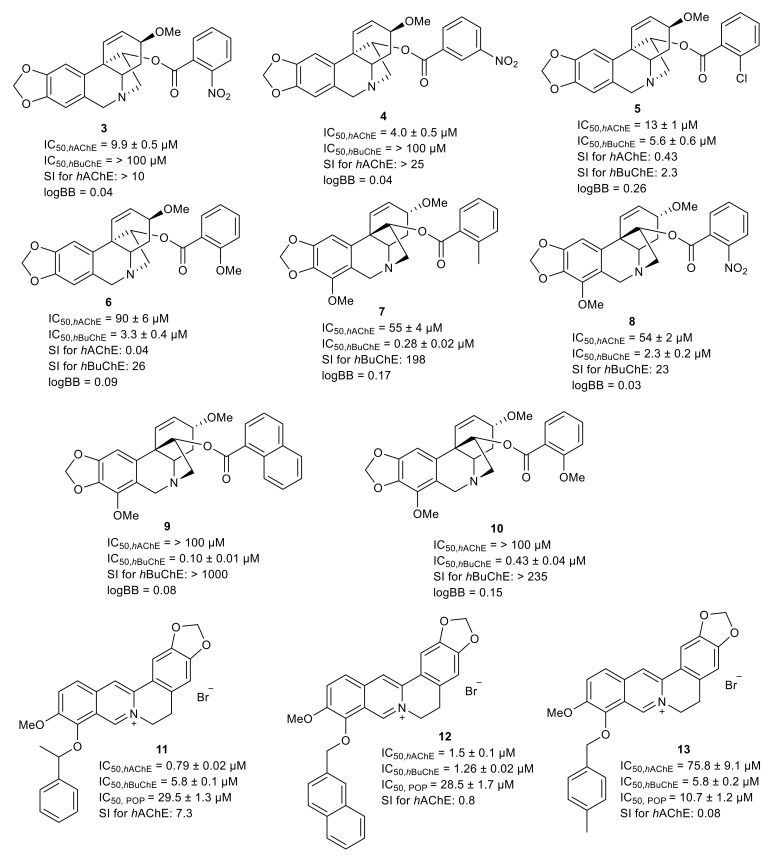
Derivatives of selected AAs and IAs together with biological activities connected with Alzheimer’s disease.

**Table 1 molecules-26-05240-t001:** Biological activities of Amaryllidaceae alkaloids, either newly isolated or tested in the last decade, in connection with Alzheimer’s disease (includes only alkaloids with an IC_50_ value ≤ 30 μM for AChE or BuChE; IC_50_ ≤ 200 μM for POP; any results for GSK-3β are included).

Alkaloid	Structural Type	Source	IC_50_ AChE (μM)	IC_50_ BuChE (μM)	IC_50_ POP (μM)	GSK-3β (μM)	Ref.
*N*-norgalanthamine	galanthamine	*Pancratium maritimum* *Lycoris longituba*	2.76 ± 0.65 ^a^ 8.85 ± 0.58 ^b^	76.4 ± 3.7 ^c^	n.s.	n.s.	[69,70]
11β-hydroxygalanthamine	galanthamine	*Lycoris longituba*	3.04 ± 0.61 ^a^	n.s.	n.s.	n.s.	[70]
narciabduliine	narcikachnine	*Narcissus pseudonarcissus* cv. Carlton	3.29 ± 0.73 ^b^	3.44 ± 0.02 ^c^	n.s.	n.s.	[64]
*N*-methylcrinasiadine	narciclassine	*Lycoris longituba*	4.23 ± 1.13 ^a^	n.s.	n.s.	n.s.	[70]
*N*-(chloromethyl)galanthamine	galanthamine	*Lycoris longituba*	5.55 ± 0.6 ^a^	n.s.	n.s.	n.s.	[70]
*O*-demethyllycoramine	galanthamine	*Lycoris longituba*	8.13 ± 1.49 ^a^	n.s.	n.s.	n.s.	[70]
deoxypretazzetine	tazettine	*Lycoris longituba*	8.44 ± 0.83 ^a^	n.s.	n.s.	n.s.	[70]
acetylcaranine	lycorine	*Amaryllis belladona*	11.7 ± 0.7 ^a^ >30 ^b^	n.s.	>200	n.s.	[65,66]
narcieliine	narcikachnine	*Zephyranthes citrina*	18.7 ± 2.3 ^b^	1.34 ± 0.31 ^c^	163 ± 13	n.s.	[63]
undulatine	crinine	*Chlidanthus fragrans* *Nerine bowdenii*	23.5 ± 1.2 ^b^	>100 ^c^	>200	43.3 ± 4.0 ^e^	[66,71,72]
8-hydroxy-9-methoxycrinine	crinine	*Pancratium maritimum*	25.3 ± 1.8 ^a^	>365 ^d^	n.s.	n.s.	[69]
*N*-(chloromethyl)lycoramine	galanthamine	*Lycoris longituba*	25.76 ± 1.09 ^a^	n.s.	n.s.	n.s.	[70]
powelline	crinine	*Nerine bowdenii*	29.1 ± 1.6 ^b^	>30 ^c^	>200	n.s.	[66]
carltonine B	belladine	*Narcissus pseudonarcissus* cv. Carlton	>30 ^b^	0.031 ± 0.001 ^c^	n.s.	n.s.	[56]
carltonine A	belladine	*Narcissus pseudonarcissus* cv. Carlton	>30 ^b^	0.913 ± 0.020 ^c^	143 ± 12	n.s.	[56]
narcimatuline	narcikachnine	*Narcissus pseudonarcissus* cv. Dutch Master	>30 ^b^	5.9 ± 0.2 ^c^	29.2 ± 1.0	20.7 ± 2.4	[62]
carltonine C	belladine	*Narcissus pseudonarcissus* cv. Carlton	>30 ^b^	14.8 ± 1.1 ^c^	n.s.	n.s.	[56]
narcipavline	narcikachnine	*Narcissus poeticus* cv. Pink Parasol	>30 ^b^	24.4 ± 1.2 ^c^	n.s.	n.s.	[61]
zephyranthine	lycorine	*Hippeastrum hybridum* cv. *Ferrari*	>30 ^b^	>30 ^c^	142 ± 10	n.s.	[73]
9-*O*-demethylgalanthine	lycorine	*Zephyranthes robusta*	>30 ^b^	>30 ^c^	150 ± 20	50.9 ± 8.9 ^e^	[74]
homolycorine	homolycorine	*Narcissus poeticus* cv. Pink Parasol	>30 ^b^	>30 ^c^	173 ± 41	54.4 ± 8.9 ^e^	[61]
nerinine	homolycorine	*Zephyranthes citrina*	>30 ^b^	>30 ^c^	190 ± 10	n.s.	[63]
masonine	homolycorine	*Narcissus poeticus* cv. Pink Parasol	>30 ^b^	>30 ^c^	>200	27.81 ± 0.05	[61,72]
caranine	lycorine	*Nerine bowdenii*	>30 ^b^	>30 ^c^	>200	30.75 ± 0.04	[66,72]
9-*O*-demethylhomolycorine	homolycorine	*Narcissus poeticus* cv. Pink Parasol	n.s.	n.s.	n.s.	30.01 ± 0.04	[72]

^a^*Ee*AChE; ^b^*h*AChE; ^c^*h*BuChE; ^d^*Eq*BuChE; ^e^ % inhibition at concentration of 50 μM HPTLC bio-autography assay.

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
