# Peer review of "Recent Progress on Biological Activity of Amaryllidaceae and Further Isoquinoline Alkaloids in Connection with Alzheimer’s Disease"

_molecules, 2021, doi:10.3390/molecules26175240_

Round 1

Reviewer 1 Report

Introduction should be more oriented towards giving the state of the art, a wider literature review is therefore essential - including isolation methods and identification of the compounds, since this is the crucial step before involving the nature derived substances into medical applications. This would also contribute to evaluate the future reader the factual significance of content and the novelty of the results.

  • Absence of graphs: All the compounds were evaluated for antioxidant and AChE inhibitory activities. It would be nice to present the antioxidant and AChE inhibitory activities in a graphical form.
  • Analytical techniques as an important step of the research in the field should be given more attention - since the extracts obtained from the plant materials are complex mixtures there is a serious obstacle regarding the separation of specific compounds like isoquinoline alkaloids.
  • I suggest to separate the sections 2 and 3 into three subsections - isolation methods, analytical methods and biological activities.
  • Discussion is very limited and the conclusions are not presented in a form that would indicate any future perspectives in the present field.

Author Response

Dear Editor,

We have carefully revised our manuscript titled “Recent progress on biological activity of Amaryllidaceae and further isoquinoline alkaloids in connection with Alzheimer´s disease " - Manuscript ID: molecules-1316422, according to the reviewers’ suggestions. All changes are highlighted in red.

Comments from the editors and reviewers:

Reviewer 1

Introduction should be more oriented towards giving the state of the art, a wider literature review is therefore essential - including isolation methods and identification of the compounds, since this is the crucial step before involving the nature derived substances into medical applications. This would also contribute to evaluate the future reader the factual significance of content and the novelty of the results.

  • Absence of graphs: All the compounds were evaluated for antioxidant and AChE inhibitory activities. It would be nice to present the antioxidant and AChE inhibitory activities in a graphical form.

We incorporated some recent information about antioxidant activity of AAs in the section dealing with neuroprotective activity of studied AAs. It is well known that antioxidant activity is connected with the presence of phenolic groups, but it has been also shown that these groups are inefficient when crossing the BBB, an important pharmacokinetic limitation for drugs which should reach the CNS area, like in AD. This phenomenon is discussed in the manuscript. Thus we didn´t incorporate the antioxidant activities of AAs into the table; anyway most of these results were obtained before 2010.

  • Analytical techniques as an important step of the research in the field should be given more attention - since the extracts obtained from the plant materials are complex mixtures there is a serious obstacle regarding the separation of specific compounds like isoquinoline alkaloids.
  • I suggest to separate the sections 2 and 3 into three subsections - isolation methods, analytical methods and biological activities.

Thank you very much for this recommendation. We have incorporated a new chapter dealing with the basic isolation methods used within the phytochemical studies of alkaloids. Moreover, we have discussed the problem of isolation artifacts.

  • Discussion is very limited and the conclusions are not presented in a form that would indicate any future perspectives in the present field.

We have improved the discussion and indicated some future perspectives of AAs and IAs for treatment of neurodegenerative diseases.

Reviewer 2

Dear Editor,

I was pleasantly surprised by the work entitled "Recent progress on biological activity of Amaryllidaceae and 2 further isoquinoline alkaloids in connection with Alzheimer´s 3 disease". In particular, the introduction is well written, concise but which includes all the facets of the disease. I found the central body of the manuscript interesting, a careful discussion of the state of the art was carried out on Amaryllidaceae alkaloids, from a chemical, pharmaceuticale and pharmacological point of view and in terms of interactions with the enzyme. Numerous derivatives have been evaluated, well characterized from the structural point of view, an excellent work. The article is well written, the only point that I think authors may improve are the conclusions, which are a little sterile compared to the amount of information provided in the manuscript, and which perhaps could be deepened, with a wider future outcome and a discussion on the more general topic. Exception for this, the article can be accepted with minor revision.

Thank you very much for this evaluation of our work. We have improved the discussion and indicated some future perspectives of AAs and IAs for treatment of neurodegenerative diseases.

Reviewer 3

Reviewer’s comments:

Authors review the most recent advances on the study of some families of Alkaloids as new drugs for Alzheimer´s disease (AD). Review covers Amaryllidaceae and related alkaloids with a main focus on their activity as anticholinesterasics, backed by the pharmacological activity of the marketed drug Galantamine. I think that this is a nice work that can serve to introduce the new students in the pharmacological study of natural products. The review is clear and concise, enriched with chemical structures and readable tables.

I have only minor points to comment that, in my opinion, would improve the review:

In some parts, the introduction seems to be comprised by bullets points. It would need a better construction. For instance, POP, MAO, and GSK3 enzymes are suddenly introduced with no sense. There is a plethora of biological targets that have been implicated in AD. But only ChE, Abeta and NMDAreceptors supply drugs to the market. For this reason, POP, MAO and GSK3 should be discussed later, where authors report the biological activities exerted by the alkaloids.

The implication of each enzyme target for AD has been improved.

I miss a better explanation of the rationale behind ChE enzymes and AD progression. I miss a paragraph about the calcium overload hypothesis, considering that memantine, a NMDA blocker, is one the few drugs prescribed for AD.

We have incorporated more detailed information about ChEs and the basics of the glutamatergic hypothesis.

In line100, paragraph should be updated to insert the recent approval of aducanumab, the first antiamyloidogenic drug prescribed for AD, and the first in the latest 18 years.

Information about this new drug has been incorpoteted into the manuscript.

In the beginning of chapter 2, authors should comment that galantamine not only exerts a ChE inhibition, but also has been described as positive allosteric modulator of neuronal nicotinic receptors (see classical contributions of Maelicke and Alburquerque’s groups, for instance).

We have added additional information about galanthamine in connection with AD.

I have only read one in vivo experiment with AAs (at the end of page 8). Authors should include more in vivo studies with the AAs, or discussing why these alkaloids have gone scarcely through to preclinical studies.

We have incorporated results from the last decade of in vivo experiments into the manuscript.

In the second paragraph of the chapter 3, authors stress the diverse ChE isoforms used for testing inhibitory activities of alkaloids. This important point should be mentioned earlier, close to the Table 1. Moreover, authors should mention that there are also several hAChEs, such as human serum AChE, human erythrocytes, or that recombined in HEK293, probably the closest to the human brain AChE. There is also isoforms used for testing anticholinesterasics from Torpedo californica (TcAChE, pacific electric ray), while that from electric eel (eeAChE) is closely related to that from human neuromuscular junction.

We have moved the section dealing with the different types of cholinesterases for the determination of inhibition activity at the beginning of section 3. The newly obtained results are discussed after this introduction paragraph.

TcAChE is often used for in silico and in vitro tests, however we have discussed results where this enzyme has not been used. We have discussed in detail especially the results where human AChE, which provides a much better model for in vitro studies, has been used for in vitro and in silico studies.

We agree that the human recombinant AChE is the best enzyme source since the method for its production is well established, and it should provide much more consistent results. However, sometimes, the exact origin of hAChE is not explicitly stated, so we decided not to discuss it in our manuscript. We also believe that the recombinant form of human AChE, especially in future years, will prevail for in vitro studies.

Finally, the conclusion needs a deeper discussion. For instance, I do not actually see a paragraph about future perspectives.

We have improved the discussion and indicated some future perspectives of AAs and IAs for treatment of neurodegenerative diseases. Moreover we have added a new chapter dealing with semisynthetic derivatives of AAs and IAs.

Further comments:

More then 30 new references have been incorporated into manuscript.

Revised manuscript has been repetitively corrected for English ny native speaker Prof. Gerald Blunden, University of Portsmouth (member of author´s team).

In light of these changes, we are positive that our revised manuscript meets the criteria to be published in Molecules and that it would be of interest for all the readers from the scientific community with particular emphasis on alkaloids and drug development against neurodegenerative disorders.

Lucie Cahlikova

Reviewer 2 Report

Dear editor,

I was pleasantly surprised by the work entitled "Recent progress on biological activity of Amaryllidaceae and 2 further isoquinoline alkaloids in connection with Alzheimer´s 3 disease". In particular, the introduction is well written, concise but which includes all the facets of the disease. I found the central body of the manuscript interesting, a careful discussion of the state of the art was carried out on Amaryllidaceae alkaloids, from a chemical, pharmaceuticale and pharmacological point of view and in terms of interactions with the enzyme. Numerous derivatives have been evaluated, well characterized from the structural point of view, an excellent work. The article is well written, the only point that I think authors may improve are the conclusions, which are a little sterile compared to the amount of information provided in the manuscript, and which perhaps could be deepened, with a wider future outcome and a discussion on the more general topic. Exception for this, the article can be accepted with minor revision.

Author Response

(The authors gave the same response as above.)

Reviewer 3 Report

Recent progress on biological activity of Amaryllidaceae and further isoquinoline alkaloids in connection with Alzheimer´s disease

Journal: Molecules; Manuscript ID 1316422

Reviewer’s comments:

Authors review the most recent advances on the study of some families of Alkaloids as new drugs for Alzheimer´s disease (AD). Review covers Amaryllidaceae and related alkaloids with a main focus on their activity as anticholinesterasics, backed by the pharmacological activity of the marketed drug Galantamine. I think that this is a nice work that can serve to introduce the new students in the pharmacological study of natural products. The review is clear and concise, enriched with chemical structures and readable tables.

I have only minor points to comment that, in my opinion, would improve the review:

In some parts, the introduction seems to be comprised by bullets points. It would need a better construction. For instance, POP, MAO, and GSK3 enzymes are suddenly introduced with no sense. There is a plethora of biological targets that have been implicated in AD. But only ChE, Abeta and NMDAreceptors supply drugs to the market. For this reason, POP, MAO and GSK3 should be discussed later, where authors report the biological activities exerted by the alkaloids.

I miss a better explanation of the rationale behind ChE enzymes and AD progression. I miss a paragraph about the calcium overload hypothesis, considering that memantine, a NMDA blocker, is one the few drugs prescribed for AD.

In line100, paragraph should be updated to insert the recent approval of aducanumab, the first antiamyloidogenic drug prescribed for AD, and the first in the latest 18 years.

In the beginning of chapter 2, authors should comment that galantamine not only exerts a ChE inhibition, but also has been described as positive allosteric modulator of neuronal nicotinic receptors (see classical contributions of Maelicke and Alburquerque’s groups, for instance).

I have only read one in vivo experiment with AAs (at the end of page 8). Authors should include more in vivo studies with the AAs, or discussing why these alkaloids have gone scarcely through to preclinical studies.

In the second paragraph of the chapter 3, authors stress the diverse ChE isoforms used for testing inhibitory activities of alkaloids. This important point should be mentioned earlier, close to the Table 1. Moreover, authors should mention that there are also several hAChEs, such as human serum AChE, human erythrocytes, or that recombined in HEK293, probably the closest to the human brain AChE. There is also isoforms used for testing anticholinesterasics from Torpedo californica (TcAChE, pacific electric ray), while that from electric eel (eeAChE) is closely related to that from human neuromuscular junction.

Finally, the conclusion needs a deeper discussion. For instance, I do not actually see a paragraph about future perspectives.

Author Response

(The authors gave the same response as above.)

Round 2

Reviewer 1 Report

The authors have adressed the remarks and the manuscript has been well implemented according to the comments.